# Development of Modern Racism Scale in Global Airlines: A Study of Asian Female Flight Attendants

**DOI:** 10.3390/ijerph18052688

**Published:** 2021-03-07

**Authors:** Myoungjin Yu, Sunghyup Sean Hyun

**Affiliations:** School of Tourism, College of Social Sciences, Hanyang University, 222 Wangsimni-ro, Seongdong-gu, Seoul 04763, Korea; rosa7767@hanyang.ac.kr

**Keywords:** global airlines, modern racism, depression, job stress, mental health, perceived insider status (PIS)

## Abstract

Due to the globalization of the airline industry, global airlines are focusing human resource management on diversity strategies and employing flight attendants of various races. Multinational flight attendants have brought many positive results; conversely, discrimination has led to negative phenomena such as racism. Nevertheless, research focusing on global airline racism in tourism studies is unprecedented. Therefore, the purpose of this study is to develop a modern racism scale rating the discrimination perceived by Asian female flight attendants on global airlines. It was developed following Churchill’s eight steps (1979). This study derived measurement items through a literature review, in-depth interviews, first and second expert surveys, and a preliminary survey. These items were developed on a scale through a validity and reliability assessment and were finally confirmed as six dimensions and 24 measurement items. Lastly, research implications were discussed.

## 1. Introduction

The globalization of the airline industry in the 2first century has made it faster and more convenient for tourists to travel abroad, and it has significantly contributed to various economic developments [1,2]. A key factor in maintaining the airline industry is passengers [3]; because they spend most of their time onboard [4], global airlines try to hire flight attendants of various races and nationalities to provide friendly service to all passengers [5]. Multinational flight attendants have brought many positive results by having opportunities to interact closely with one another; conversely, ignoring their diversity has led to negative phenomena, such as racism. Despite efforts to prevent racism, subtle modern racism is happening without difficulty [6,7,8]. Many scholars suggest that racism in these global organizations can have a negative impact on employees’ physical and mental health as well as their job satisfaction [9,10,11,12].

Corita (2008) mentioned that the subtle modern racism in the global workplace is much more frequent than the overt, old-fashioned one; however, most work-related studies have focused on preventing and imposing old-fashioned racism, and they are in an early stage in the understanding of modern racism [6]. In particular, because most Northeast Asians do not have any experience or general knowledge of racism, they have more difficulties in recognizing subtle modern racism: only after the situation has unfolded are they able to realize that it was an implication of racism [13]. Nevertheless, research focusing on global airline racism in tourism studies is unprecedented. Therefore, the purpose of this study is to (1) develop a scale of modern racism for Asian female flight attendants by applying the intersectionality theory, and (2) confirm its theoretical validity through a correlation by which the theoretically related gold standard variables are empirically relevant for the newly developed modern racism scale.

## 2. Conceptual Framework

### 2.1. Global Airline

According to the World Tourism Organization (WTO), the growth of international tourists and the expansion of the tourism industry are closely related to that of the global airline industry, which has been growing at a steady rate of 57% over the past 10 years, from 2008 to 2017, compared to the 37% of roads, 4% of sea, and 2% of railways. The proportion of passengers using global airlines has continued to grow from 33.5% in 2010 to 35.5% in 2016, strengthening the airlines’ competitiveness, which was revealed through a variety of strategies, including aggressive marketing, in-flight service upgrading, strategic alliances among airlines, and customer service preferential programs [14]. In particular, airlines employ multiracial and multinational flight attendants to improve the quality of their in-flight service [5]. For example, many airlines in the Americas, Europe, and the Middle East—including United, Delta, British Airways, Northwest Airlines, Lufthansa, Swissair, KLM Royal Dutch Airlines, Finnair, AIRFRANCE, Emirates, Qatar Airways, Kuwait Airways, Gulf Air, and Saudia—employ crews of various races and nationalities [15].

Flight attendants working for multiracial and multinational global airlines closely interact with one another through dormitory life on overseas bases, narrow in-flight and bunker environments, and by sharing hotels and tourist activities while abroad. This working environment produces positive results, such as strengthening competitiveness through creativity and productivity; however, ignoring diversity has led to negative phenomena, such as racism.

### 2.2. Modern Racism

Cacas (2005) defined racism as an act of maintaining a belief that a particular race is physically, psychologically, and intellectually superior compared to other races by recognizing race as a biological being [16]. Essed (1990) defined racism as an act of treating people unfairly and justifying it by classifying a particular ethnic group into an apparently inferior group [17].

Old-fashioned, traditional racism was based on a belief system by which Black people should be discriminated against because White people are superior in the relationships between them. However, after World War II, institutional mechanisms for restricting old-fashioned racism were established, and as people’s perceptions changed, traditional forms of racism gradually faded away [13]. However, racism has not disappeared, but it has become a new and modern form of racism that is more subtle, clever, and subversive compared to past clear and public forms [18]. Though racism has now acquired a different form, it remains a critical issue, and many aspects must still be improved [19]. Modern racism can be classified into five main categories: symbolic racism, aversive racism, color-blind racism, everyday racism, and racial microaggressions.

### 2.3. Intersectionality Theory

The early discussion of the intersectionality theory is historically based on the strategies and insights of the racial resistance movement of various women including the Sojourner Truth abolitionists and the Combahee River Collective in Black women’s lesbian organizations [20]. The intersectionality theory was first used in a 1989 paper by Black American lawyer Kimberlé Crenshaw as a conceptual framework to simultaneously study racism and sexism toward Black women suffering from double oppression in the United States [21].

Crenshaw argued that if an individual standing at an intersection had a traffic accident, it could be caused by many cars coming from different directions. Thus, if Black women were discriminated against, it could be a product of both racism and sexism [21]. Therefore, social inequality and discrimination in intersection recognition have become a useful conceptual tool for understanding differences among people—especially women in different positions—by delivering a multi-discrimination message by which individuals can define their location and identity as results of complexity, such as race, gender, nationality, class, age, religion, and so on [22]. Similarly, Collins (2002) also studied the dominant interlinked matrix of the racial discrimination, gender discrimination, and class discrimination systems. The intersectionality theory provided a broad understanding of inequality and discrimination in post-colonial states [23,24].

### 2.4. Gold Standard Variables

#### 2.4.1. Depression

Recently, depression has not been viewed as pathological but rather as part of the adaptation process to universal human situations, and its experience may vary depending on personal and environmental characteristics [25]. As a result, researchers who understand that depression is a symptom of social relationships rather than a pathological disease are conducting studies looking at the social factors that affect it [26].

Many studies have shown that perceived discrimination in social relationships, such as racism, is closely related to increased depressive symptoms [27,28,29]. Based on these studies, it is believed that the modern racism perceived by flight attendants on global airlines is also related to an increase in their depression.

#### 2.4.2. Work Stress 

Blau, Tatum, McCoy, Dobria, and Ward-Cook (2004) defined work stress as a conflict between employees and the organizational environment [30]. It has already been mentioned in many studies that racism (experienced or perceived) can have negative consequences on life [31,32,33,34,35]; the study above showed that modern racism experiences in the workplace can be quite harmful to employees, as individuals are publicly discriminated against, which can affect work stress [7]. Racism stress in the workplace for women of ethnic minorities has shown more negative effects than for men [36,37,38,39]. Based on these studies, it is believed that modern racism perceived by flight attendants on global airlines is also related to increasing their work stress.

#### 2.4.3. Mental Health

Wetzler (1989) described mental health as an emotional state of well-being, that is, the ability to resolve conflicts based on rational decisions and to continuously respond to environmental stress and internal pressure [40]. The first study on the relationship between mental health and perceived discrimination began with researchers interested in racism. Since the publication of a paper on the effects of racism, sexism, and class discrimination on mental health by Kreiger and his colleagues in 1993, related papers have continuously been published [26].

Various studies proved that the modern racism experiences of Asian Americans increased psychological pain [41,42,43] and diminished mental health [44,45,46,47]. Anderson (1991) also suggested that racism could cause mental health-related problems for individual victims [48,49]. Such perceived discrimination, especially perceived racism, is an extremely negative issue for mental health in general [50]. Based on these studies, it is believed that modern racism perceived by flight attendants on global airlines has a harmful effect on their mental health.

#### 2.4.4. Perceived Insider Status

The perceived insider status (PIS) first proposed by Stamper and Masterson indicates the level at which an individual recognizes an organization differently or is accepted as an insider within the workplace. Masterson and Stamper (2002) said that PIS could represent the sense of belonging that employees are aware of [51]. The experience of being ignored, discriminated against, and attacked by bosses or colleagues in the workplace will make the employee reevaluate the relationship with the organization and be perceived as an outsider rather than an insider as the PIS decreases [52,53]. Based on these studies, it is believed that modern racism perceived by flight attendants on global airlines has a harmful impact on PIS. 

## 3. Scale Development Procedure

This study aims to improve the level of racism perception through the development of a modern racism scale reflecting the perception of Asian female flight attendants through an inductive approach. Among the various scale development procedures presented through prior studies [54], the modern racism scale will be based on the eight-step scale development procedure of Churchill (1979). The scale development procedures in this study are shown in Table 1.

### 3.1. Analysis through Literature Review

The first step is the analysis step through a literature review. Items can be collected through the literature analysis related to prior research, and they can be changed to better meet the purpose of the study during reconstruction [55].

To derive measurement items of modern racism on global airlines, initial measurement items were derived through a prior literature review related to modern racism for Asian employees in the global workplace [56,57,58,59,60]. These measurement items were reviewed by two experts who have worked for more than a decade at global airlines to ensure content validity and surface validity for the measurement items through a link between the theoretical background and empirical process.

### 3.2. Analysis through Qualitative Research of In-Depth Interviews

The second step is the analysis through qualitative research of in-depth interviews. This study conducted in-depth interviews with former and current Northeast Asian female flight attendants. The characteristics of participants in in-depth interviews are shown in Table 2.

### 3.3. Collect First Data

It is the process of collecting the first initial items through a literature review and conducting the qualitative research of in-depth interviews.

### 3.4. Refining Measurement Items (First and Second Expert Survey)

Refining collected measurement items refers to procedures for ensuring validity, assessing whether items are applicable as a compositional concept through the review and evaluation of experts in the relevant fields. Content validity is the determination of whether the items being collected reflect the constructive concepts that are intended to be measured [61]; after the items are reviewed by experts, they are evaluated for their appropriateness to explain constructive concepts [62]. 

In terms of expert surveys, these should be composed considering the suitability, representation, sincerity, and professionalism of the experts, and the ideal number of experts is ideally judged by 5–20 people. Based on this, the study selected 10 Northeast Asian female flight attendants living in the UK, Spain, Dubai, and South Korea. The status of the first and second expert surveys are shown in Table 3. The expert survey method first contacted the experts individually and explained the research topics, objectives, and details of the survey in advance. The survey was distributed through e-mail or social networking services, and experts could freely describe items by moving, modifying, integrating, deleting, and adding categories through a five-point Likert scale (1 point = “not at all”, 5 points = “very much”). The results of the expert surveys were obtained by calculating the average, standard deviation, content validity, and refining matters through Excel and SPSS 23.

Based on this first expert survey, the second expert investigation also derived the average, standard deviation, content validity, and refining matters, and it recorded the details of adoption, deletion, modification, integration, movement, and additional items.

### 3.5. Collect Second Data 

The literature review, in-depth interviews, and the first and second expert surveys have led to the formation of secondary final items of modern racism on global airlines.

### 3.6. Validity and Reliability Assessment of Scale 

The survey was conducted after a pre-test to see if the initial configuration items derived from the literature review, in-depth interviews, and expert surveys were applicable as a modern racism scale on global airlines. The preliminary survey was conducted for five days (4–8 August 2020) through online preliminary surveys provided to 20 people, including tourism professors, researchers, external instructors, and PhD students as well as former and current flight attendants.

As a result of the preliminary survey analysis, the secondary preliminary survey was conducted for three days (10–12 August 2020) to ensure internal validity after a comprehensive and final modification, category shift, deletion, and addition of items. Exploratory factor analysis, confirmative factor analysis, and Cronbach’s alpha coefficients using SPSS 23 and AMOS 23 were employed to evaluate the validity and reliability of the data collected from the main survey. Finally, a scale on modern racism in global airlines was developed. 

### 3.7. Theoretical Validity Evaluation for Scale Verification

For the newly developed scale verification, the criterion and nomological validity should be checked. First, the newly developed scale can be verified as a general scale through the verification process of the criterion validity [63]. To measure it, it is necessary to establish the gold standard variables—the measurement variables that represent the criteria—and to ensure that the newly developed scale and the gold standard variables correlate. That is, the correlation coefficient or validity coefficient represents the criterion validity, where the range of the coefficient ranges from 0.00 (which indicates no relationship between the two measurements) to 1.00 (which indicates that the two measurements are completely correlated) [63]. 

Next, nomological validity indicates that if there is a theoretical relationship between different constructions, the corresponding relationship between the measured values is confirmed [64]. Based on a logical theory or prior study, they’re predicted the relationship between two constructive concepts in a positive direction and judged that it has nomological validity when the relationship between the real latent variables appears to be a significantly positive one. However, even if the relationship between the latent variables is negative (−) or positive (+), if the actual relationship between the real latent variables is not statistically significant, we judge that there is no law justification. Thus, the nomological validity refers to the direction between the latent variables, and we can check the correlation to see whether the relationship between the variables is positive (+) or negative (−) and significant [65].

This study conducted a correlation analysis between the newly developed modern racism variables and the gold standard variables (depression, work stress, mental health, and PIS) to check the criterion and nomological validity of the newly developed modern racism scale on global airlines.

### 3.8. Scale Development

A new scale of modern racism on global airlines was developed after checking the verification and evaluation.

## 4. Scale Development Results

### 4.1. Development of Initial Measurement Items

Through a literature review, in-depth interviews, first and second expert surveys, and preliminary surveys, items were divided into seven preliminary dimensions and 35 measurement items, including three items of inferiority in ability, three items of negative overestimation, four items of appearance inferiority, four items of identity loss, eight items of passive character, seven items of work unfairness, and six items of nonmainstream recognition.

### 4.2. Development of Final Dimensions and Measurement Items

After completing the second preliminary surveys, the newly developed measurement items were surveyed online for 24 days, from 13 August to 5 September 2020, by former and current Northeast Asian female flight attendants of seven global airlines to reflect the work experiences of various ethnic flight attendants in the Middle East, Europe, and Asia. To find the study subjects, it was searched on a social media platform such as Instagram using hashtags. After confirming the study subjects, the online google surveys were conducted by introducing the researcher, the purpose of the research, and a questionnaire through individual direct messages. The questionnaire is shown in Appendix A in detail. In addition, exploratory factor analysis, reliability analysis, and confirmative factor analysis were conducted using SPSS 23 and AMOS 23 to ensure validity and reliability.

#### 4.2.1. Demographic Characteristics

Table 4 shows the results of the demographic characteristics by frequency analysis of 230 out of 245 respondents; 15 respondents who answered with duplicate numbers were excluded.

First, the age distribution of the respondents was 47.0% (108 people) in their 30s, 42.2% (97 people) in their 20s, and 10.9% (25 people) in their 40s. Among the respondents, 86.1% (198 people) were South Korean, about 5.7% (13 people) were Japanese, and 8.3% (19 people) were Chinese. Most of them were unmarried (80.4%; 185 people), with 19.6% (45 people) married, while the rest had a different status. In addition, 99.6% (229 people, all but one) had a college degree or higher. The percent of flight attendants working the three Middle East airlines was 89.1% (205 people), followed by 5.7% (13 people) of the three European airlines, and 5.2% (12 people) of one Asian airline. The total sample had an even distribution of current flight attendants (58.7%; 135 people) and former flight attendants (41.3%; 95 people).

In terms of position distribution, economy flight attendants were the most with 58.7% (135 people), business flight attendants were 21.7% (50 people), first flight attendants were 10% (23 people), assistant pursers were 8.7% (20 people), and pursers were 0.9% (two people). As for the working period, the majority (41.3%; 95 people) had worked for less than 2 years, 26.5% (61 people) between 2 and 5 years, and 27.4% (63 people) between 5 and 10 years.

The overseas period was the highest for 32.2% of the sample (74 people), with a period ranging between 5 and 10 years, followed by 29.6% (68 people) between 2 and 5 years, 20% (46 people) with less than 2 years, 12.2% (28 people) with a period ranging between 10 and 15 years, 4.3% (10 people) between 15 and 20 years, and 1.7% (four people) with more than 20 years. 

#### 4.2.2. Dimensionality Verification through Exploratory Factor Analysis

Measurement items in this study model have eliminated some items through the scale purification process. For validity verification, this study conducted an exploratory factor analysis of seven preliminary named dimensions and 35 items, including inferiority in ability, negative overestimation, appearance inferiority, identity loss, passive character, work unfairness, and nonmainstream recognition.

If the eigenvalue was 1.0 or higher, and the factor loading value was 0.4 or higher during the analysis process, the variable was considered acceptable. However, if the factor loading values exhibited 0.4 or higher simultaneously in two or more dimensions, which do not conform to the theoretical structure of the previous study, the corresponding item was removed. In this analysis, a total of three items were eliminated, and seven dimensions and 32 items were used, as shown in Table 5. 

As a result of the final analysis, the Kaiser–Meyer–Olkin (KMO) measure was high at 0.906, and Bartlett’s spherical test showed 4807.218 (p < 0.001) of $\chi^{2}$ to confirm that the data were suitable for factor analysis. In addition, the seven dimensions were extracted as the same theoretical structure of the previous study results, with a total variance of 68.743% explained. These dimensions were named “nonmainstream recognition”, “passive character”, “appearance inferiority”, “negative overestimation”, “work unfairness”, “inferiority in ability”, and “identity loss”.

Moreover, to analyze the reliability of the seven dimensions extracted, the Cronbach’s alpha value was checked. Overall, the dimensions were judged to be reliable, with a value of 0.6 or higher, including 0.939 in the “nonmainstream recognition”, 0.904 in the “passive character”, 0.815 in the “appearance inferiority”, 0.858 in the “negative overestimation”, 0.842 in the “work unfairness”, 0.802 in the “inferiority in ability”, and 0.622 in the “identity loss”—although the last one was found to be slightly lower.

#### 4.2.3. Confirmatory Factor Analysis

##### Model Fit

As a result of conducting the confirmatory factor analysis to confirm the validity of the compositional concepts for the seven dimensions and 32 items obtained from the exploratory factor analysis (EFA), 10 problem items were derived from squared multiple correlations with a value of 0.5 or less and excessive observation error values. Among them, eight items were deleted, and two items were recognized as essential elements during the scale development; the final analysis was conducted again for 24 items. At this time, all four items of the identity loss dimension that showed low reliability values in the EFA were deleted and refined into six dimensions.

As such, the fit of the final model (six dimensions, 24 items) was determined as $\chi^{2}$ = 402.866, p < 0.001, $\chi^{2}$/d.f = 1.752, GFI = 0.876, NFI = 0.903, RFI = 0.883, TLI = 0.946, CFI = 0.9557, and RMSEA = 0.057, and it was judged that the overall goodness of fit criterion was satisfied.

##### Convergent Validity

To validate the convergent validity of the refined measurement model (6 dimensions, 24 items), the following values were identified for standardized factor loads (λ) of 0.5 or higher, average variance extracted (AVE) of 0.5 or higher, and construct reliability (CR) of 0.7 or higher. As the results all met the standards, they were judged to have convergent validity, as shown in Table 5.

##### Discriminant Validity

To verify the discriminant validity of the construct, the requirements had to confirm that the AVE value of each latent variable (dimension) was greater than the square of the correlation coefficient ($\mathrm{AVE}>\rho^{2}$), and it was asserted that 1 should not be included in the confidence interval of the correlation coefficient showing the correlation between the latent variables (ρ±2×S·E≠1). To ensure whether $\mathrm{AVE}>\rho^{2}$ of the first condition is satisfied, the correlation between each latent variable was first checked; the results are shown in Table 6.

As shown in the table above, the correlation coefficient (ρ) between “passive character” and “negative overestimation” is 0.760, and the square of the correlation coefficient ($\rho^{2}$) is 0.578, indicating the largest value. Further, the AVE value for “passive character” is 0.589, and the AVE value for “negative overestimation” is 0.641; thus, it is evident that it is greater than the square of the correlation coefficients between the two latent variables. Therefore, it can be inferred that the AVE value of all latent variables is greater than the square of the correlation coefficients between each latent variable.

Next, it was checked whether the second condition was satisfied, namely that 1 should not be included in the range, by adding or subtracting the value obtained by multiplying the standard error by 2 in the correlation coefficient. As a result, all values represent values between 0.070 and 0.902, and the range confirmed that 1 was not included in it (ρ±2×S·E≠1).

In conclusion, both the above conditions were met, and we were able to confirm that the study model has discriminant validity.

#### 4.2.4. Final Factor Analysis

The final loading values and Cronbach’s α for each dimension were reconfirmed for the six dimensions and 24 items that were determined through this study. The results showed that the KMO measures were 0.894, and $\chi^{2}$ was 3987.119 (p < 0.001) in Bartlett’s spherical test, confirming that the data were suitable for factor analysis and that the total variance explained by the six dimensions was 75.454%. In addition, the Cronbach’s α values for the six dimensions showed an overall reliability of 0.7 or higher, with 0.939 for “nonmainstream recognition”, 0.902 for “passive character”, 0.855 for “appearance inferiority”, 0.858 for “negative overestimation”, 0.834 for “work unfairness”, and 0.802 for “inferiority in ability”.

### 4.3. Verification of the Developed Scale

The proposed final research hypotheses and research model for the newly developed scale evaluation are as follows in Figure 1.

#### 4.3.1. Final Research Hypothesis

**H1.** 
*Nonmainstream recognition will have a positive (+) significant effect on depression.*


**H2.** 
*Nonmainstream recognition will have a positive (+) significant effect on work stress.*


**H3.** 
*Nonmainstream recognition will have a positive (+) significant effect on mental health.*


**H4.** 
*Nonmainstream recognition will have a negative (−) significant effect on PIS.*


**H5.** 
*Passive character will have a positive (+) significant effect on depression.*


**H6.** 
*Passive character will have a positive (+) significant effect on work stress.*


**H7.** 
*Passive character will have a positive (+) significant effect on mental health.*


**H8.** 
*Passive character will have a negative (−) significant effect on PIS.*


**H9.** 
*Appearance inferiority will have a positive (+) significant effect on depression.*


**H10.** 
*Appearance inferiority will have a positive (+) significant effect on work stress.*


**H11.** 
*Appearance inferiority will have a positive (+) significant effect on mental health.*


**H12.** 
*Appearance inferiority will have a negative (−) significant effect on PIS.*


**H13.** 
*Negative overestimation will have a positive (+) significant effect on depression.*


**H14.** 
*Negative overestimation will have a positive (+) significant effect on work stress.*


**H15.** 
*Negative overestimation will have a positive (+) significant effect on mental health.*


**H16.** 
*Negative overestimation will have a negative (−) significant effect on PIS.*


**H17.** 
*Work unfairness will have a positive (+) significant effect on depression.*


**H18.** 
*Work unfairness will have a positive (+) significant effect on work stress.*


**H19.** 
*Work unfairness will have a positive (+) significant effect on mental health.*


**H20.** 
*Work unfairness will have a negative (−) significant effect on PIS.*


**H21.** 
*Inferiority in ability will have a positive (+) significant effect on depression.*


**H22.** 
*Inferiority in ability will have a positive (+) significant effect on work stress.*


**H23.** 
*Inferiority in ability will have a positive (+) significant effect on mental health.*


**H24.** 
*Inferiority in ability will have a negative (−) significant effect on PIS.*


#### 4.3.2. Final Research Model

#### 4.3.3. Correlation Analysis

Criterion validity and nomological validity (theoretical validity) were used to verify whether it can be used as a general scale of the six latent variables (24 measured variables) derived from the new scale of modern racism. For this validity, 245 former and current Northeast Asian female flight attendants of global airlines were surveyed online for 24 days between 13 August and 5 September 2020.

The dimensions of the survey were divided into six parts, including modern racism, depression, work stress, mental health, PIS, and demographic characteristics. The gold standard dimensions consisted of eight items on depression [66], seven items on work stress [67], seven items on mental health [68], and three items on PIS [69].

The table below confirms the correlation between variables by conducting a Pearson correlation analysis to determine the relative influence between the six dimensions of modern racism and the four dimensions of gold standard.

As shown in Table 7, it was confirmed that all six dimensions (①–⑥) of modern racism had a positive (+) correlation with the gold standard variables of depression, work stress, and mental health, while five dimensions (②–⑥) of modern racism had a negative (−) correlation with the gold standard variable of PIS. 

Among the positive (+) correlations, the correlation between nonmainstream recognition and depression was the highest, and that between appearance inferiority and depression was the lowest. Furthermore, among the negative (−) correlations, the correlation between appearance inferiority and PIS was the highest, and that between inferiority in ability and PIS was the lowest. 

#### 4.3.4. Final Scale Development Results

The scale of modern racism perceived by Asian flight attendants on global airlines, which was finally completed after confirming the verification stage of the new scale, is shown in Table 8. It shows six items on the recognition of nonmainstream Asian flight attendants, six items on the passive character of Asian female flight attendants, three items on the appearance inferiority of Asian female flight attendants, three items on the negative overestimation of Asian flight attendants, three items on the work unfairness of Asian flight attendants, and three items on the inferior ability of Asian flight attendants, comprising a total of six dimensions and 24 measurement items in Table 9.

## 5. Discussion 

This study derived measurement items through a literature review, in-depth interviews, first and second expert surveys, and preliminary surveys to develop a modern racism scale as perceived by Asian female flight attendants on global airlines.

These measurement items were developed on a scale through a validity and reliability assessment and were finally confirmed as six dimensions and 24 measurement items. The six dimensions extracted, all regarding aspects of Asian female flight attendants, were named “nonmainstream recognition”, “passive character”, “appearance inferiority”, “negative overestimation”, “work unfairness”, and “inferior ability”. In particular, it can be seen that the dimensions of passive character and appearance inferiority represent the dual oppression and discrimination against Asian women because they’re a minority and female at the same time, like the intersectionality theory mentioned by Kimberlé Crenshaw [21].

Airline flight attendants are much more often women, but they also have the characteristics of a profession that provides services with strict appearance standards. Therefore, it was found that Asian female flight attendants have a more specific and diverse perception of appearance racism, which was poorly treated as ‘Asians all look alike’ in previous studies [59,60,61]. On the other hand, through several previous studies, it was found many items of Asian’s non-mainstream recognition and Asian’s ability inferiority. However, most of them were not adopted as the measurement items of modern racism of Asian flight attendants. Through this, it can be judged that the non-mainstream recognition and the ability inferiority of Asian flight attendants are perceived less seriously than other employees in global workplaces, or are gradually improving due to the time gap with previous studies. However, it should not be overlooked the seriousness of those which still exist as factors of modern racism.

To verify the criterion and nomological validity of the newly developed scale, we analyzed the correlation between the modern racism latent variables and the gold standard latent variables, including depression, work stress, mental health, and PIS. As a result, it was confirmed that all six dimensions of the newly developed scale were positively correlated with depression, work stress, and mental health; and five dimensions—excluding nonmainstream recognition—were confirmed to have a negative correlation with PIS.

Through this result, the newly developed scale measuring modern racism in global airlines was generally proven to have a significant correlation with the gold standard variables of depression, work stress, mental health, and PIS. It suggests that Asian female flight attendants are negatively psychologically and emotionally affected by the experiences of modern racism, which increase depression, work stress, and mental health, and are perceived by airlines as outsiders rather than insiders.

## 6. Implications

The academic implications of this study are as follows. First, although necessary, the focus on global airline racism in tourism research is unprecedented. Therefore, this study is the first of its kind and provides a theoretical basis for various future studies of related researchers. Second, it is of great significance because it presents the social phenomenon of racism as a scale equipped with specific dimensions and items that can be quantitatively measured. Third, its significance is also revealed by the fact that a modern racism scale was developed by applying the intersectionality theory and considering the specificity of women as well as the ethnic minorities of Asian female flight attendants.

The practical implications of this study are as follows. First, providing future education on racism in global airlines will have a positive effect on preventing racial discrimination and improving awareness. Second, this study will help protect victims of racism by actively developing systems and policies that counter subtle modern racism as well as obvious old-fashioned racism in global airlines. Third, those managing crew team composition should review the schedule arrangements more thoroughly to ensure that flight attendants do not perceive themselves as minorities during the flight. In addition, the purser and assistant purser will need to identify the flight attendants of the minority race in briefings, flights, and layovers and manage them more thoughtfully. Fourth, a study by Kim Eun-ha (2018) found that, when it comes to gender discrimination, women are more aware of appearance discrimination than men [70]. It is perceived as more difficult for flight attendants in global airlines who are also ethnic and female to become promotional models for the airline; moreover, they feel that there are significantly more White female models. This phenomenon conveys a message of multiple discrimination through the intersectionality theory, with sexism and racism applied simultaneously. This will require global airlines to further consider the balanced distribution of various races in selecting promotional model crews. It will also be necessary for the main characters to be played by women of various races in airline advertising.

Fifth, Asian flight attendants were found to be aware of unfairness in their work as they continued to respond to passengers’ call bells before flight attendants of other races. Accordingly, the purser and assistant purser managing the flight attendants will need to make strategic efforts to fairly distribute the service work among them. Lastly, in addition to working with multiracial people, global flight attendants receive considerably good pay and welfare benefits, and the working period was found to be considerably short, ranging from one to two years [71]. In this study, we can consider that depression, work stress, mental health, and the perception of crew members as outsiders are affected by negative organizational cultures, such as racism in airlines. Therefore, great efforts should be made to improve organizational culture through network group activities for mentoring, social support, sharing information [72], and participation in programs to improve relationships among multiracial flight attendants.

## 7. Limitation and Recommendation for Future Research

The limitations for the generalization of this study and future research directions are presented as follows. First, there was a limit to collecting sufficient surveys due to the characteristics of being former and current Northeast Asian female flight attendants who worked with various races. Second, surveys were collected from the former and current flight attendants of three airlines in the Middle East, three in Europe, and one in Asia; however, there was a limitation in the uneven distribution of the world’s work area due to the difficulty of collecting surveys in the Americas. For the generalization of future studies, it will be necessary to ensure that the subjects are more evenly distributed globally. Third, although this study was conducted on Northeast Asian flight attendants in Japan, China, and South Korea, most respondents are from South Korea due to the limitation in collecting surveys from Japanese and Chinese flight attendants. In future studies, it will be necessary to ensure that the nationalities of the subjects are evenly distributed. Fourth, as it was an initial study on racism in the airline field, and not enough previous related research could be presented as a theoretical basis, the current study shows a limit to derive abundant measurement items. We expect future relevant research to be more diverse based on this study’s theoretical presentation. Fifth, through in-depth interviews, expert surveys, and main surveys, individual differences in modern racism as perceived by Asian female flight attendants could be detected, though with limitations in finding the cause. Future research is expected to examine such individual differences in-depth and concretely suggest ways to overcome modern racism.

## Figures and Tables

**Figure 1 ijerph-18-02688-f001:**
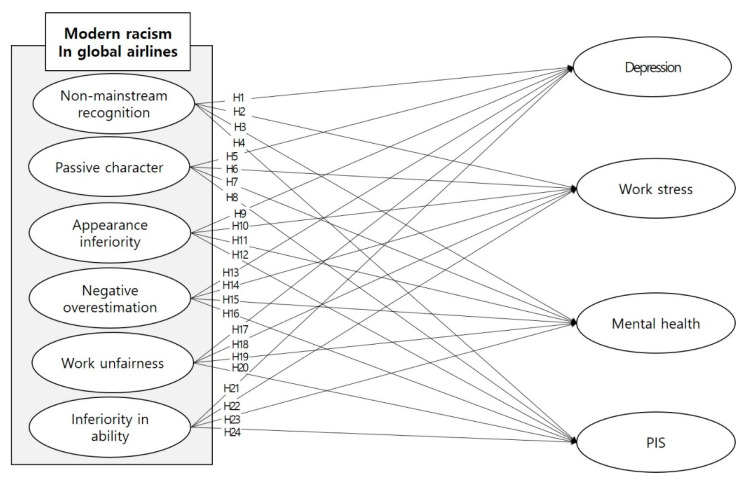
The research model of modern racism.

**Table 1 ijerph-18-02688-t001:** Procedures of scale development in this study.

Step	Process for Developing Modern Racism Scale on Global Airlines	
1	Analysis through literature review	Development of composition concepts through a literature review on modern racism in the workplace.
2	Analysis through qualitative research of in-depth interviews	Development of composition concepts through in-depth interviews with Asian female flight attendants.
3	Collect first data	The collection stage of the first initial items through analysis of literature and qualitative research.
4	Refine measurement items	(1) First expert survey for content validity assessment and refining. (2) Second expert survey and refining based on the first expert survey.
5	Collect second data	Collection stage of the second final items for modern racism on global airlines.
6	Validity and reliability assessment of scale	Validity and reliability assessment steps for developing derived items on a scale.
7	Validity assessment for new scale verification	Theoretical validity assessment steps with scale validation of newly developed scales.
8	Scale development	Development of a final scale after checking the scale verification step.

**Table 2 ijerph-18-02688-t002:** Characteristics of in-depth interviews with participants.

	ID	Nationality	Career	Position	Education
1	A	China	13 years	Purser	University graduate
2	B	China	7 years	First class	University graduate
3	C	Japan	6 years	First class	University graduate
4	D	Japan	7 years	First class	University graduate
5	E	South Korea	10 years	Purser	Graduate school graduate
6	F	South Korea	11 years	Assistant purser	Graduate school graduate
7	G	South Korea	10 years	Assistant purser	University graduate
8	H	South Korea	5 years	First class	Graduate school graduate
9	I	South Korea	4 years	Business class	Graduate school graduate

**Table 3 ijerph-18-02688-t003:** Status of first and second expert surveys.

	First Expert Survey	Second Expert Survey
Survey period	6–13 July 2020	20–28 July 2020
Survey method	Online	Online
Experts	10 people	10 people
Analysis method	Average Standard deviation Content validity ratio	Average Standard deviation Content validity ratio

**Table 4 ijerph-18-02688-t004:** Demographic characteristics.

Index		No.	%	Index		No.	%
Age	20–29	97	42.2	Working status	Former	95	41.3
	30–39	108	47.0		Current	135	58.7
	Over 40	25	10.9	Working position	Purser	2	0.9
Nationality	Japan	13	5.7		Assistant purser	20	8.7
	China	19	8.3		First class	23	10.0
	South Korea	198	86.1		Business class	50	21.7
Marital status	Married	43	18.7		Economy class	135	58.7
	Single	185	80.4	Working period	Under 2 years	95	41.3
	Etc.	2	0.9		2–5 years under	61	26.5
Education level	High school	1	0.4		5–10 years under	63	27.4
	College	203	88.3		Over 10 years	11	4.8
	In graduate school	10	4.3	Overseas period	Under 2 years	46	20.0
	Graduate degree	16	7.0		2–5 years under	68	29.6
Working area	Asia	12	5.2		5–10 years under	74	32.2
	Europe	13	5.7		10–15 years under	28	12.2
	Mideast	205	89.1		15–20 years under	10	4.3
-	-	-	-		Over 20 years	4	1.7

**Table 5 ijerph-18-02688-t005:** Exploratory factor analysis and reliability analysis results.

Dimensions	Measurement Items	Factor Loading	Eigenvalue	Variance exp. p (%)	Cronbach’s Alpha
Nonmainstream recognition	I felt a small presence during briefing.	0.862	5.001	15.628	0.939
	I felt social alienation during briefing.	0.862			
	I felt a small presence during flight work.	0.844			
	I felt social alienation during flight work.	0.809			
	I felt a small presence during layover time.	0.805			
	I felt social alienation during layover time.	0.767			
Passive character	People think Asian women are obedient.	0.763	4.425	13.827	0.904
	People think Asian women are passive.	0.716			
	People think most Asian women agree with their superior’s words.	0.702			
	I’ve heard that I am different from other Northeast Asian women when I express my opinion strongly.	0.695			
	People think Asian women struggle to actively assert their opinions.	0.680			
	People think Asian women have difficulty expressing resoluteness.	0.639			
	People think Asian women are silent when problems arise.	0.622			
Appearance inferiority	I’ve felt that the appearance of White women is preferred in an airline.	0.829	3.232	10.099	0.815
	It is more difficult for Asian women to be selected as an airline’s promotional models.	0.814			
	Promotion models have significantly more White female flight attendants.	0.790			
	Most main roles in airline ads are played by White women.	0.643			
	Most of the senior executives seem to be of a different race from me.	0.557			
Negative overestimation	People think Asians’ sincerity is natural.	0.770	2.599	8.122	0.858
	People think Asians’ diligence is natural.	0.738			
	People think it is natural for Asians to work harder.	0.710			
Work unfairness	I felt unfair treatment from superiors of other races.	0.714	2.511	7.846	0.842
	I felt unfair treatment from colleagues of other races.	0.701			
	I felt I worked more than colleagues of other races.	0.576			
	People expect Asians to immediately respond to passengers’ requests.	0.575			
Inferiority in ability	People have a prejudice that Asians cannot speak English well.	0.785	2.232	6.976	0.802
	People think Asians lack communication skills.	0.750			
	I felt pressured to continuously prove my abilities and values.	0.645			
Identity loss	I’ve heard from people of other races that Northeast Asian look similar.	0.655	1.998	6.243	0.622
	I’ve been asked for answers to social issues in other Asian countries because I’m Asian.	0.563			
	I felt that other colleagues want me to respond to Asian passenger.	0.552			
	A flight attendant of my race seems to be more difficult to get the office management job.	0.447			
Kaiser–Meyer–Olkin measure of sampling adequacy = 0.906					
Bartlett’s test of sphericity		$\chi^{2}$		4807.218	
		Df (p)		496 (0.000)	
Total variance explained by 7 factors: 68.743%					

**Table 6 ijerph-18-02688-t006:** Confirmative factor analysis results.

Dimensions	Measurement Items	Standardization Coefficient (*λ*)	*AVE*	*CR*
Nonmainstream recognition	I felt a small presence during briefing.	0.859	0.638	0.913
	I felt social alienation during briefing.	0.834		
	I felt a small presence during flight work.	0.889		
	I felt social alienation during flight work.	0.886		
	I felt a small presence during layover time.	0.789		
	I felt social alienation during layover time.	0.754		
Passive character	People think Asian women are obedient.	0.814	0.589	0.896
	People think Asian women are passive.	0.755		
	People think most Asian women agree with their superior’s words.	0.822		
	People think Asian women struggle to actively assert their opinions.	0.727		
	People think Asian women have difficulty expressing resoluteness.	0.732		
	People think Asian women are silent when problems arise.	0.735		
Appearance inferiority	It is more difficult for Asian women to be selected as an airline’s promotional models.	0.777	0.636	0.840
	Promotion models have significantly more White female flight attendants.	0.861		
	Most main roles in airline ads are played by White women.	0.817		
Negative overestimation	People think Asians’ sincerity is natural.	0.764	0.641	0.842
	People think Asians’ diligence is natural.	0.867		
	People think it is natural for Asians to work harder.	0.841		
Work unfairness	I felt unfair treatment from superiors of other races.	0.884	0.606	0.818
	I felt unfair treatment from colleagues of other races.	0.915		
	People expect Asians to immediately respond to passengers’ requests.	0.614		
Inferiority in ability	People have a prejudice that Asians cannot speak English well.	0.790	0.530	0.769
	People think Asians lack communication skills.	0.888		
	I felt pressured to continuously prove my abilities and values.	0.634		

**Table 7 ijerph-18-02688-t007:** Correlation results between latent variables.

Dimensions	*AVE*	*CR*	1	2	3	4	5	6
1. Nonmainstream recognition	0.638	0.913	1					
2. Passive character	0.589	0.896	0.490 (0.240)	1				
3. Appearance inferiority	0.636	0.840	0.204 (0.042)	0.443 (0.196)	1			
4. Negative overestimation	0.641	0.842	0.379 (0.144)	0.760 (0.578)	0.386 (0.149)	1		
5. Work unfairness	0.606	0.818	0.619 (0.383)	0.633 (0.401)	0.273 (0.075)	0.514 (0.264)	1	
6. Inferiority in ability	0.530	0.769	0.405 (0.164)	0.612 (0.375)	0.403 (0.162)	0.487 (0.237)	0.454 (0.206)	1

**Table 8 ijerph-18-02688-t008:** Correlation analysis results.

Variables	①	②	③	④	⑤	⑥
Depression	0.715 **	0.299 **	0.132 *	0.233 **	0.413 **	0.345 **
Work stress	0.374 **	0.236 **	0.175 **	0.218 **	0.283 **	0.289 **
Mental health	0.457 **	0.246 **	0.198 **	0.197 **	0.286 **	0.271 **
PIS	−0.114	−0.146 *	−0.251 **	−0.153 *	−0.187 **	−0.135 *

Note (1): ① Nonmainstream recognition, ② Passive character, ③ Appearance inferiority, ④ Negative overestimation, ⑤ Work unfairness, ⑥ Inferiority in ability. Note (2): * *p* < 0.05, ** *p* < 0.01.

**Table 9 ijerph-18-02688-t009:** Modern racism scale in global airlines.

	Dimensions	Measurement Items	Cronbach’s α
1	Nonmainstream recognition	I felt a small presence during briefing.	0.939
		I felt social alienation during briefing.	
		I felt a small presence during flight work.	
		I felt social alienation during flight work.	
		I felt a small presence during layover time.	
		I felt social alienation during layover time.	
2	Passive character	People think Asian women are obedient.	0.902
		People think Asian women are passive.	
		People think most Asian women agree with their superior’s words.	
		People think Asian women struggle to actively assert their opinions.	
		People think Asian women have difficulty expressing resoluteness.	
		People think Asian women are silent when problems arise.	
3	Appearance inferiority	It is more difficult for Asian women to be selected as an airline’s promotional models.	0.855
		Airline promotions have significantly more White female flight attendants.	
		Most main flight attendant roles in airline ads are played by White women.	
4	Negative overestimation	People think Asians’ sincerity is natural.	0.858
		People think Asians’ diligence is natural.	
		People think it is natural for Asians to work harder.	
5	Work unfairness	I felt unfair treatment from superiors of other races.	0.834
		I felt unfair treatment from colleagues of other races.	
		People expect Asians to immediately respond to passengers’ requests.	
6	Inferiority in ability	People have a prejudice that Asians cannot speak English well.	0.802
		People think Asians lack communication skills.	
		I felt pressured to continuously prove my abilities and values.	

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
