# Peer review of "Development of Modern Racism Scale in Global Airlines: A Study of Asian Female Flight Attendants"

_ijerph, 2021, doi:10.3390/ijerph18052688_

Round 1
Reviewer 1 Report
Very interesting and relevant research, also due to the very well presented theoretical basis and the deep and extensive analysis of empirical data collected from different resources. Perhaps only the conclusions could be more generalized, but in any case, they are clearly based on empirical data. Thus, they are grounded and relevant to theory and practice. It is important to emphasize that even the limits of the research are very clear, as well as ideas for future research in such an important area where, according to the author (s) of the article, there are many very subtle forms of modern racism and, in my view, many hidden ways of discriminating against the most vulnerable groups in society.
Author Response
Dear Reviewer
We are pleased to submit our revised manuscript entitled "Development of Modern Racism Scale in Global Airlines: Focused on Asian Female Flight Attendants" to the international Journal of Environmental Research and Public Health ( Manuscript ID: IJERPH - 1096229). "Please see the attachment for 1) ERB Exemption Myoungjin Yu, 2) the revised manuscript."
We are thankful to you and the referees for reviewing this manuscript. We have revised this manuscript based on the comments. We hope the revision has improved the manuscript to your satisfaction.
In addition, we have made professional communication and linguistic presentation a priority in the revision. Editorial errors (e.g., grammar, punctuation, verb tense, spelling) were thoroughly rechecked, and sentence structure and overall writing throughout the manuscript were substantially improved.
The authors are identified and described on the title page, but not within the manuscript document. The paper has not been published and is not under consideration at any other publication.
Thank you very much for the opportunity to submit our manuscript.
We do appreciate your assistance and consideration.
Sincerely,
Sunghyup Sean Hyun, Ph. D.
Associate Professor
School of Tourism, Hanyang University
17 Haengdang-dong, Seongdonggu
Seoul 133-791, Republic of Korea
Phone: 82-2-2220-0862
Fax: 82-2-2281-4554
E-mail: sshyun@hanyang.ac.kr
Summary of Revisions for Manuscript ID# – IJERPH 1096229
Manuscript title:
Development of Modern Racism Scale in Global Airlines : Focused on Asian Female Flight Attendants
We sincerely appreciate the reviewers’ comments and suggestions for the previous version of this manuscript. We have thoroughly studied them and have revised the manuscript accordingly. This report summarizes our responses to all the comments.
===================================================================
Reviewer 1
Comments to the Author
Very interesting and relevant research, also due to the very well presented theoretical basis and the deep and extensive analysis of empirical data collected from different resources. Perhaps only the conclusions could be more generalized, but in any case, they are clearly based on empirical data. Thus, they are grounded and relevant to theory and practice. It is important to emphasize that even the limits of the research are very clear, as well as ideas for future research in such an important area where, according to the author (s) of the article, there are many very subtle forms of modern racism and, in my view, many hidden ways of discriminating against the most vulnerable groups in society.
Thank you very much for this valuable comment. We applied your comment in the revised manuscript by improving the part of 5. Discussion. Please see the paragraphs below.
=> 5. Discussion
These measurement items were developed on a scale through a validity and reliability assessment and were finally confirmed as six dimensions and 24 measurement items. The six dimensions extracted, all regarding aspects of Asian female flight attendants, were named “nonmainstream recognition,” “passive character,” “appearance inferiority,” “negative overestimation,” “work unfairness,” and “inferior ability.”
These measurement items were developed on a scale through a validity and reliability assessment and were finally confirmed as six dimensions and 24 measurement items. The six dimensions extracted, all regarding aspects of Asian female flight attendants, were named “nonmainstream recognition,” “passive character,” “appearance inferiority,” “negative overestimation,” “work unfairness,” and “inferior ability.” In particular, it can be seen that the dimensions of passive character and appearance inferiority represent the dual oppression and discrimination against Asian women because they’re a minority and female at the same time, like the intersectionality theory mentioned by Kimberlé Crenshaw [21].
Airline flight attendants are not only made up of much more women, but also have the characteristics of a profession that provides services with strict appearance standards. Therefore, it was found that Asian female flight attendants have a more specific and diverse perception of appearance racism, which was poorly treated as 'Asians all look alike' in previous studies [59, 60, 61]. On the other hand, through several previous studies, it was found many items of Asian's non-mainstream recognition and Asian's ability inferiority. But most of them were not adopted as the measurement items of modern racism of Asian flight attendants. Through this, it can be judged that the non-mainstream recognition and the ability inferiority of Asian flight attendants are perceived less seriously than other employees in global workplaces, or are gradually improving due to the time gap with previous studies. However, it shouldn’t be overlooked the seriousness of those which still exist as factors of modern racism.
We truly appreciate your time and help. Your comments and suggestions definitely have improved the quality of this manuscript.
===================================================================
Reviewer 2
Comments to the Author
1. Some of the keywords seem inappropriately chosen. What is the relevance of keywords such as “Depression”, “Job stress” and “Mental health”?
Thank you very much for this precious question.
This study was developed a modern racism scale (6 factors and 24 measuring variables) reflecting the perception of Asian female flight attendants. And to verify whether the new scale can be used as a general scale, the theoretical validity of the criterion validity and nomological validity were confirmed. To check it, it is necessary to establish the gold standard variables that represent the criteria. And to ensure that the gold standard variables and the new scale should be theoretically correlated.
In this study, in order to test the theoretical validity of the new scale, gold standard variables were set as variables theoretically related to modern racism: depression, job stress, mental health, and perceived insider status. Then the criterion validity and nomological validity were confirmed through the correlation analysis between new scale and gold standard variables.
The above was mentioned in section ‘3.7. Theoretical validity evaluation for scale verification’ of the paper.
2. The introduction could be more succinct. Information such as “As of 2017, Alexandre de ~ 2017 [3]” is superfluous and unnecessary.
Thank you very much for your valuable comments.
Based on your review, this part was deemed unnecessary and has been deleted from the introduction. And we’re reorganized the number of references.
3. "Did the study obtain institutional review board (IRB) approval? Please provide the actual IRB study/approval number.
Thank you very much for this keen observation.
When we designed this research, we requested the IRB(institutional review board) approval in our university. IRB committee reviewed our research during June in 2020. They decided that our research falls within “IRB exempt status”. Based on the IRB committee’s decision, we could start actual survey to participants. We translated the "Letter of approval from the Ethics Review Committee" into English. This approval letter was issued in June 2020 when we conducted the initial research. Please find the attached file (File name: ERB Exemption Myoungjin Yu).
4. How were the participants recruited? This was unclear.
Thank you very much for this keen observation.
To find the study subjects, it was searched on social media using hashtags. After confirming the study subjects, the online surveys were conducted by introducing the researcher, the purpose of the research, and a questionnaire through individual direct messages.
The above was added in section ‘4.2. Development of final dimensions and measurement items’ of the paper. Please see the paragraph below.
=> 4.2. Development of final dimensions and measurement items
After completing the re-preliminary surveys, the newly developed measurement items were surveyed online for 24 days, from August 13 to September 5, 2020, by former and current Northeast Asian female flight attendants of seven global airlines to reflect the work experiences of various ethnic flight attendants in the Middle East, Europe, and Asia.
After completing the re-preliminary surveys, the newly developed measurement items were surveyed online for 24 days, from August 13 to September 5, 2020, by former and current Northeast Asian female flight attendants of seven global airlines to reflect the work experiences of various ethnic flight attendants in the Middle East, Europe, and Asia. To find the study subjects, it was searched on social media using hashtags. After confirming the study subjects, the online surveys were conducted by introducing the researcher, the purpose of the research, and a questionnaire through individual direct messages.
5. ‘...15 respondents were excluded who were unfaithful“-What does this mean? Please clarify.
Thank you very much for your precious question.
15 unfaithful (unscrupulous) respondents mean the people responding with the same number on survey.
The above was added in section ‘4.2.1. Demographic characteristics’ of the paper. Please see the paragraph below.
=> 4.2.1. Demographic characteristics
Table 4 shows the results of the demographic characteristics by frequency analysis of 230 out of 245 respondents;
Table 4 shows the results of the demographic characteristics by frequency analysis of 230 out of 245 respondents; 15 unscrupulous respondents, such as responding with the same number, were excluded.
6. Were the interviews conducted in English or was translation avaliable? I am not sure what is meant by “I felt a small presence”. Is this readily understood by participants and readers?
Thank you very much for this constructive question.
230 surveys were composed in Korean and English. Of these, 198 answered the questionnaire in Korean and 32 answered the questionnaire in English. None of the participants asked about the meaning of small presence. So I judged that there was no difficulty in understanding the small presence.
In this study, small presence means a state in which it is difficult to receive positive attention from the organization and colleagues.
7. Do you mean “resolve” rather than “resoluteness”?
Thank you very much for this precious question.
Resolve means like determination or will, but resoluteness has a similar meaning to firm or assertive. Therefore, it is judged that resoluteness is more appropriate than resolve in this study.
8. Serveral compendiums of instruments that measure perceived racism and/or discrimination are present in the literature. Other works have reviewed the psychometric properties of these instruments in terms of validity and reliability and have indicated if the instrument was factor analyzed. However, the authors have paid little attention to the quality of the factor analysis performed.
Thank you very much for this constructive comment.
We have added Table 5 to supplement the factor analysis explanation in section ‘4.2.2. Dimensionality verification through exploratory factor analysis in this paper.
Thank you so much for your comments and the time you gave for the improvement of our manuscript.
===================================================================
Reviewer 3
Comments to the Author
This manuscript by Myoungjin Yu et al is wholistic and detailed. In this study, a modern racism scale is developed to evaluate the discrimination perceived by Asian flight attendants on global airlines. Literature review, in-depth reviews and confirmatory factor analysis are described in sufficient detail. The correlation analysis is well described and presented. 'Gold standard variables' described in this work including depression, work stress, mental health, perceived insider status are critical factors from the standpoint of public health and are explained sufficiently well in this work. The reviewer believes that the developed scale in this work is novel and the review holds merit to be published in IJERPH.
Thank you so much for your positive comments and the time you gave for our manuscript.

Reviewer 2 Report
This study tried to develop a modern racism scale rating the discrimination perceived by Asian female flight attendants on global airlines. Although novel, the research design and methods lack adequate description. Extensive editing of the English language and style is required as well.
Specific comments:
- Some of the keywords seem inappropriately chosen. What is the relevance of keywords such as "Depression", "Job stress" and "Mental health"?
- The introduction could be more succinct. Information such as "As of 2017, Alexandre de Juniac, CEO of the International Air Transport Association (IATA), predicted that in 20 years the number of global air passengers would be double the four billion passengers of 2017 [3]" is superfluous and unnecessary.
- Did the study obtain institutional review board (IRB) approval? Please provide the actual IRB study/approval number.
- How were the participants recruited? This was unclear.
- "... 15 respondents were excluded who are unfaithful" - what does this mean? Please clarify.
- Were the interviews conducted in English or was translation available? I am not sure what is meant by "I felt a small presence". Is this readily understood by participants and readers?
- Do you mean "resolve" rather than "resoluteness"?
- Several compendiums of instruments that measure perceived racism and/or discrimination are present in the literature. Other works have reviewed the psychometric properties of these instruments in terms of validity and reliability and have indicated if the instrument was factor analyzed. However, the authors have paid little attention to the quality of the factor analysis performed.
Author Response

(The authors gave the same response as above.)

Reviewer 3 Report
This manuscript by Myoungjin Yu et al is wholistic and detailed. In this study, a modern racism scale is developed to evaluate the discrimination perceived by Asian flight attendants on global airlines. Literature review, in-depth reviews and confirmatory factor analysis are described in sufficient detail. The correlation analysis is well described and presented. 'Gold standard variables' described in this work including depression, work stress, mental health, perceived insider status are critical factors from the standpoint of public health and are explained sufficiently well in this work. The reviewer believes that the developed scale in this work is novel and the review holds merit to be published in IJERPH.
Author Response

(The authors gave the same response as above.)

Round 2
Reviewer 2 Report
Some corrections to methodology and text editing still required.
Specific comments:
- Please provide the ERC approval number in the manuscript. There should still be a unique study/approval number regardless.
- The surveys were conducted online - but over which platform exactly. Please specify.
- "15 unscrupulous respondents" - is "unscrupulous" the right word choice here? Should it just be "duplicate"?
- The underlying data should be made publicly available. If this was not possible, please provide a reason why.
Author Response
Dear Reviewer 2
We are pleased to submit our revised manuscript entitled "Development of Modern Racism Scale in Global Airlines: Focused on Asian Female Flight Attendants" to the international Journal of Environmental Research and Public Health (Manuscript ID: IJERPH - 1096229). "Please see the attachment for the revised manuscript."
We are thankful to you and the referees for reviewing this manuscript. We have revised this manuscript based on your 2nd comments. We hope the revision has improved the manuscript to your satisfaction.
In addition, we have made professional communication and linguistic presentation a priority in the revision. Editorial errors (e.g., grammar, punctuation, verb tense, spelling) were thoroughly rechecked, and sentence structure and overall writing throughout the manuscript were substantially improved.
The authors are identified and described on the title page, but not within the manuscript document. The paper has not been published and is not under consideration at any other publication.
Thank you very much for the opportunity to submit our manuscript.
We do appreciate your assistance and consideration.
Sincerely,
Sunghyup Sean Hyun, Ph. D.
Associate Professor
School of Tourism, Hanyang University
17 Haengdang-dong, Seongdonggu
Seoul 133-791, Republic of Korea
Phone: 82-2-2220-0862
Fax: 82-2-2281-4554
E-mail: sshyun@hanyang.ac.kr
Summary of Revisions for Manuscript ID# – IJERPH 1096229
Manuscript title:
Development of Modern Racism Scale in Global Airlines : Focused on Asian Female Flight Attendants
We sincerely appreciate the reviewers’ comments and suggestions for the previous version of this manuscript. We have thoroughly studied them and have revised the manuscript accordingly. This report summarizes our responses to all the comments.
=================================================
Reviewer 2
Comments to the Author
1. Please provide the ERC approval number in the manuscript. There should still be a unique study/approval number regardless.
Dear Reviewer, thank you very much for your time and comments. As we attached the ERB exemption decision letter at the 1st round of review, the ethics committee decided that “this study falls within the exempt status”. In other words, this study does not need IRB approval. The decision was “this study does not need approval”: How do we get the “approval number”?
2. The surveys were conducted online - but over which platform exactly. Please specify.
Thank you very much for this keen observation. This study questionnaire was collected through Instagram using Google surveys. Based on your review, this part has been supplemented in section ‘4.2. Development of final dimensions and measurement items’ as follows.
=> 4.2. Development of final dimensions and measurement items
To find the study subjects, it was searched on social media using hashtags. After confirming the study subjects, the online surveys were conducted by introducing the researcher, the purpose of the research, and a questionnaire through individual direct messages.
To find the study subjects, it was searched on a social media platform such as Instagram using hashtags. After confirming the study subjects, the online Google surveys were conducted by introducing the researcher, the purpose of the research, and a questionnaire through individual direct messages on Instagram.
3. "15 unscrupulous respondents" - is "unscrupulous" the right word choice here? Should it just be "duplicate"?
Thank you very much for your valuable question. Based on your review, This part has been modified in section ‘4.2.1. Demographic characteristics’ as follows.
=>4.2.1. Demographic characteristics
Table 4 shows the results of the demographic characteristics by frequency analysis of 230 out of 245 respondents; 15 unscrupulous respondents, such as responding with the same number, were excluded.
Table 4 shows the results of the demographic characteristics by frequency analysis of 230 out of 245 respondents; 15 respondents who answered with duplicate numbers were excluded.
4. The underlying data should be made publicly available. If this was not possible, please provide a reason why.
Thank you very much for the valuable suggestion. We send (1) SPSS data set and (2) survey questionnaire. First, We attached the survey questionnaire as an Appendix in the manuscript. Second, we send the SPSS data set to the editor, so it can be made publicly available. You can also use the data for future study. Please don’t hesitate to contact the editor if you have any question regarding of this. I appreciate your valuable opinion once again.
